

# Effects of plant age on antioxidant activity and endogenous hormones in Alpine *Elymus sibiricus* of the Tibetan Plateau

Juan Qi[1], Zhaolin Wu[1], Yanjun Liu[1] and Xiangjun Meng[2]

[1] Key Laboratory of Grassland Ecosystem of Ministry of Education, College of Grassland Science, Gansu Agricultural University, Lanzhou, Gansu Province, China
[2] Gansu Grassland Technical Extension Station, Lanzhou, Gansu Province, China

Corresponding author
Juan Qi, qijuan0622@163.com

## ABSTRACT

*Elymus sibiricus* L. is a perennial forage species that has potential to serve as a forage source in livestock grazing systems. However, *E. sibiricus* has been shown to have a rapid and substantial reduction of aboveground biomass and seed yield after 3 or 4 years and an accelerated aging process. To determine possible aging mechanisms, we planted *E. sibiricus* seeds in triplicate blocks in 2012, 2015, and 2016, respectively, and harvested samples of leaves and roots at the jointing and heading stages in 2018 and 2019 to determine oxidative indices and endogenous hormones. The fresh aboveground biomass of 4- and 5-year old plants declined by 34.2% and 52.4% respectively compared with 3-year old plants, and the seed yield declined by 12.7% and 34.1%, respectively. The water content in leaves was 51.7%, 43.3%, and 35.6%, and net photosynthesis was 7.73, 6.35, and 2.08 μmol/m²·s in 3-, 4-, and 5-year old plants, respectively. The superoxide anion radical generation rate in leaves and roots did not show any aging pattern. There was a non-significant increase in malondialdehyde concentration with plant age, particularly in leaves and roots at the heading stage in 2019. The superoxide dismutase activity showed a declining trend with age of plant roots at the jointing stage in both 2018 and 2019. The peroxidase activity declined with plant age in both leaves and roots, for example, and the catalase activity in roots 4- and 7-year old plants declined by 13.8% and 0.85%, respectively, compared to 3-year old plants at the heading stage in 2018. Therefore, the reduced capacity of the antioxidant system may lead to oxidative stress during plant aging process. Overall, the concentrations of plant hormones, auxin (IAA), gibberellin (GA), zeatin (ZT), and abscisic acid (ABA) were significantly lower in roots than in leaves. The IAA concentration in leaves and roots exhibited different patterns with plant age. The ZT concentrations in leaves of 3-year old plants was 2.39- and 2.62-fold of those in 4- and 7-year old plants, respectively at the jointing stage, and in roots, the concentration declined with plant age. The changes in the GA concentration with plant age varied between the physiological stages and between years. The ABA concentrations appeared to increase with plant age, particularly in leaves. In conclusion, the aging process of *E. sibiricus* was apparently associated with an increase in oxidative stress, a decrease of ZT and an increase of ABA, particularly in roots. These findings highlight the effects of plant age on the antioxidant and endogenous hormone activity of *E. sibiricus*. However, these plant age-related trends showed variations between plant physiological stages and between harvest years that

needs to be researched in the future to develop strategies to manage this forage species.

# INTRODUCTION

Perennial herb *Elymus sibiricus* L. belongs to *Poaceae* genus and is an important species in the alpine region of the Tibetan Plateau and the steppe region of northern Eurasia (*Ma et al., 2009*; *Xiong et al., 2021*). *E. sibiricus* has many prominent characteristics that can be used for these grasslands and restoration of deteriorated grasslands for livestock production (*Yan et al., 2007*; *Ma et al., 2008*). However, *E. sibiricus* population in grasslands is vulnerable to decline and plant yields are reduced with increasing plant age, which prevents its use for long-term pastures over large areas (*Jin, 2021*). Recent studies indicate that physiological burdens, such as the changes of phytohormones in the aging process, age-induced oxidative stress and age-related changes in water relations and photosynthesis are responsible for reduced plant growth with age (*Munné-Bosch & Lalueza, 2007*). It is hypothesized that with increasing age of *E. sibiricus* plants there will be intrinsic changes in biochemistry and hormone metabolism that will affect the plant physiology leading to a productivity reduction. Therefore, studying the occurrence of aging-related physiological mechanism and regulation in *E. sibiricus* has an important theoretical and practical significance that can be applied to forage plant maintenance.

Plant aging is a highly complex process influenced by plant metabolism, as well as environmental factors (*Munné-Bosch & Lalueza, 2007*). Many mechanisms for plant aging have been proposed, such as nutrient deficiencies, excessive free radicals related to the aging process, and plant hormones changes (*Ashok & Ali, 1999*; *Jibran, Hunter & Dijkwel, 2013*; *Kraj, 2016*). The excessive accumulation of free radicals and the disturbance of endogenous hormone profile in the cells can cause oxidant stress and a deterioration of plant growth and metabolism (*Ashok & Ali, 1999*; *Rustin et al., 2000*; *Chen et al., 2020*). The reactive oxygen species (ROS) are by-products of many metabolic processes, and ROS accumulation is a key feature of plant senescence (*Ashok & Ali, 1999*). With the onset of plant senescence, ROS such as superoxide anion radical ($O_2^{\bullet-}$) and other free radicals are excessively produced (*Jing et al., 2008*) resulting in peroxidation of membrane lipids, damage to macro molecules, and even programmed cell death (*Breusegem & Dat, 2006*).

Plants have a variety of defense strategies, such as antioxidant enzymes and non-enzymatic antioxidants, to cope with ROS stress (*Shri et al., 2009*; *Farooq et al., 2015*). Superoxide dismutase (SOD) is the first line of antioxidant enzymes to scavenge ROS by converting $O_2^{\bullet-}$ to oxygen ($O_2$) and $H_2O_2$. Then, $H_2O_2$ is reduced rapidly to $H_2O$ and $O_2$ catalyzed by catalase (CAT) or peroxidase (POD) (*Noodén, Guiamet & John, 1997*; *Palma et al., 2006*). Malondialdehyde (MDA) is a primary end-product of lipid peroxidation in plants, and its concentration is usually used to indicate the severity of oxidative stress. For example, plant tissues under abiotic stress have an increase in MDA content (*Duan*

*et al., 2014*; *Suzuki et al., 2012*). The role of oxidative stress in plant senescence and aging has been demonstrated especially in annual and biennial species (*Quirino, Normanly & Amasino, 1999*) but this situation is less clear in longer lived plant species.

Previous research has also shown that endogenous hormone content and their balance plays an essential role in the aging process of perennials (*Munné-Bosch & Lalueza, 2007*; *Guo et al., 2022*). Ethylene and abscisic acid (ABA) are recognized as key hormones in plant aging, and these play critical role in the regulation of growth and development of the entire life cycle of plants (*Zwack & Rashotte, 2015*; *Niu et al., 2015*). Stress-induced ethylene and ABA production is involved in the generation of ROS (*Surplus et al., 1998*; *Orozco-Cardenas & Ryan, 1999*; *Pellinen, Palva & Kangasjarvi, 1999*; *Zakari et al., 2020*). It has been shown that endogenous ABA concentration in 7-year-old *Cistus clusii* plants was higher than in 2-year-old (*Munné-Bosch & Lalueza, 2007*). In this case, 2-year-old plants had already reached maturity, in which the growth rate was reducing (*Finkelstein, Gampala & Rock, 2002*). In contrast, retardants include cytokinin (CK), auxin (IAA), gibberellin (GA) and their related compounds have been shown to delay plant aging (*Saniewski et al., 2020*). The ability of newly emerged leaves to produce auxins and cytokinins declined with plant age in conifers. This outcome supports a relationship between reduced growth and auxin and cytokinins decreases in aging perennials (*Valdés, Fernández & Centeno, 2004*; *Aldés, Centeno & Fernández, 2004*). Research has shown that these phytohormones do not work alone, but and they are often functioning concomitantly to regulate plant senescence. (*Noodén & Leopold, 1988*). Currently, researches on the age-related hormone regulation and mechanisms of perennial plants focus mainly on trees (*Chen et al., 2020*) and annual plants such as *Arabidopsis*, maize, and rice (*Cui et al., 2020*; *Xiao et al., 2020*; *Zakari et al., 2020*), and little information is available on the hormonal changes in various tissues of perennial grasses during plant aging process.

In this study, we used *Elymus sibiricus* cv. Qingmu No. 1 (a novel variety) planted in the Tibet Plateau region for 3, 4, 5, 7, and 8 years and determined the changes in the antioxidant system and endogenous hormones in leaves and roots at jointing and heading stages to investigate their roles in plant aging.

## MATERIALS AND METHODS

### Field site

The experiment was carried out at the Haiyan Research Station of Qinghai Province, China (E100°85′, N36°45′) from June to October 2018 and repeated in 2019. The averaged altitude of the location is 3,159 m and the mean annual temperature is 0.6 °C. The average annual precipitation is about 369–403 mm, mostly occurring during the plant's growing season (July to September). The average annual evaporation is 1,435 mm, the sunshine duration is 2,985 h, and the frost-free period lasts approximately 30 days. The monthly rainfall and mean temperature in 2018 and 2019 is shown in Fig. 1. Changes of temperature and rainfall are synchronized throughout the year, and the warm season is from July to September with higher rainfall. The soil is the chernozem soil type, and the chemical properties of soil are shown in Table 1.

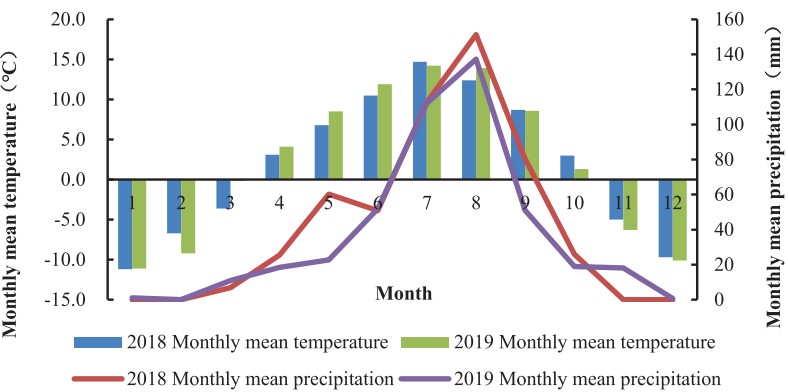

**Figure 1** Monthly mean temperature and cumulative rainfall at the Haiyan Research Station, Haiyan County, Qinghai Province, China.

**Table 1 Chemical properties of soil at Haiyan Research Station, Haiyan County, Qinghai Province, China.**

| pH | OM % | Total N g/kg | Total K g/kg | Total P g/kg | Available nitrogen mg/kg | Available potassium mg/kg | Available phosphorus mg/kg |
|---|---|---|---|---|---|---|---|
| 8 | 2.93 | 1.22 | 7.73 | 0.41 | 68.19 | 213.87 | 10.53 |

## Experimental design and field management

*E. sibiricus* seeds were planted in 2012, 2015 and 2016, respectively with three replicate plots each year. *E. sibiricus* seeds were obtained from Alpine Meadow Ecosystem Research Station of Qinghai Academy of Animal Science and Veterinary Medicine, China. This variety was selected from *Elymus* population in the Lake Area of Qinghai Province and is suitable for planting in regions with an altitude of 4,500 m. Plot size was 4 m × 5 m with 0.5 m corridors between the plots. Seeds were sown in rows with 30 cm space between rows, sowing depth was 3 cm, and the seeding rate was 4.5 g/m$^2$. Fertilizer (urea 10 g·m$^{-2}$ and superphosphate 50 g·m$^{-2}$)was applied before sowing, and with no further fertilizer applications after sowing. Plant growth was dependent on rainfall and weeds were manually removed from the plots throughout the experimental periods. In 2018 and 2019, leave and root samples of plants were collected in late June (in the jointing stage of the grasses) and late July (in the heading stage of the grasses) respectively for measurement and analyses with the plant materials aged 3, 4, 5, 7, and 8 years being available for analysis (Table 2).

## Samples

One hundred growth tiller branches at the jointing (24 June 2018 and 22 June 2019) and heading growth stages (22 July 2018 and 24 July 2019), respectively, were randomly selected in each plot. The top three leaves of each branch with similar shapes were collected. The leaves were immediately separated from stems. Five 30 × 30 cm square sections in each plot with uniform growth were randomly selected and the roots within the

**Table 2 Plant age of *E. sibiricus* samples at harvest.**

| Years of sowing | Plant age | |
| --- | --- | --- |
| | 2018 | 2019 |
| 2016 | 3 | 4 |
| 2015 | 4 | 5 |
| 2012 | 7 | 8 |

first 20 cm of soil were fully extracted and cleaned in running water. All samples were immediately snap-frozen in liquid nitrogen and stored at −80 °C for subsequent analyses.

The aboveground biomass in the anthesis period was harvested from 1 m$^2$ quadrat for each plot during 2019. Samples were weighed (fresh weight) and then dried in an oven at 65 °C for 48 h to determine dry matter content (dry weight) (*Yan et al., 2007*). The RWC was determined using the equation: RWC = 100 [(FW − DW)/(TW − DW)], where FW is the fresh weight, TW is the turgid weight, and DW is the dry weight, respectively (*Zhang, 2003*). At seed maturity, the reproductive branches were removed from a randomly selected 1 m$^2$ quadrat in each plot. After air drying, seeds were threshed, weighed, and used to calculate seed yield (kg/ha).

## Sample processing and assays
### SPAD-chlorophyll and net photosynthesis (Pn)
Ten plants at the anthesis stage in each plot were randomly selected to measure the chlorophyll content and net photosynthesis (Pn) in each of the selected flag leaves. The SPAD-chlorophyll values were recorded using a portable SPAD-502 Chlorophyll Meter (Osaka, Japan). The Pn value was recorded using a LI-6400/XT Portable Photosynthesis System (LI-COR Biosciences, Lincoln, NE, USA) from 09:00 to 11:00 on a sunny day. The readings for all selected flag leaves in each plot were used to calculate means for SPAD-chlorophyll and Pn.

## Determination of superoxide anion radical
The superoxide anion radical ($O_2^{\bullet-}$) generation rate for leaf and root samples was determined using the hydroxylamine oxidation method as described by *Hao, Kang & Yu (2007)* with modifications. A total of 1 g of leaves and roots were ground with a mortar and pestle in 3 mL of 50 mmol/L potassium phosphate buffer (pH 7.8) solutions. The reaction mixture comprised of 0.5 mL of the extracts, 0.5 mL of 50 mmol/L potassium phosphate (pH 7.8) buffer and 1 mL of 10 mmol/L hydroxylamine was incubated at 30 °C for 1 h. Subsequently, 1 mL of 17 mmol/L sulfanilic acid (water preparation) and l mL of 7 mmol/L α-naphthylamine was added, and the mixture was maintained at 30 °C for 30 min. Then, $O_2^{\bullet-}$ concentration was determined at 530 nm against a calibration curve with known concentrations of nitrite as the standard.

## Determination of MDA concentration

The concentration of MDA was measured according to the method of *Qiu et al. (2008)* with modifications. Briefly, 0.5 g frozen leaf or root was homogenized in 10 mL of phosphate buffer (pH: 7.8) in an ice bath and centrifuged at $15,000 \times g$ and $4\,°C$ for 20 min. One milliliter of supernatant was mixed with 2 mL of 0.6% thiobarbituric acid solution, incubated at $95\,°C$ in a water bath for 15 min, then quickly cooled to room temperature for 2 min and the mixture was centrifuged at $5,000 \times g$ and $25\,°C$ for 10 min. The absorbance of the solution was determined at 450, 532, and 600 nm ($A_{450}$, $A_{532}$ and $A_{600}$) respectively using UV-2450 spectrophotometer (Shimadzu, Japan).

## Determination of antioxidant enzyme activities

To determine antioxidant enzymes activities, 0.5 g frozen leaves or roots were grounded with a mortar and pestle in liquid N, and added 2 mL of phosphate buffer (0.05 mol/L, pH 7.8, a mixture of $Na_2HPO_4$ and $NaH_2PO_4$). The mixture was centrifuged at $11,000 \times g$ and $4\,°C$ for 20 min and the supernatant was used to determine the activities of antioxidant enzymes.

## SOD activity

For the SOD activity, 100 µL supernatant was added into 4 mL of the reaction mixture that consisted of 2 mL of 0.05 mol/L phosphate buffer, 0.5 mL of 104 mmol/L methionine, 1 mL of 300 µmol/L nitroblue tetrazolium, and 0.5 mL of 0.3 mmol/L disodium ethylenediaminetetraacetic acid (EDTA-$Na_2$). The solution was placed under 4,000 Lux/Watt fluorescent lamps for 10 min and the absorbance was recorded at 560 nm.

## CAT activity

The CAT activity was determined using the method as described by *Zhang (2003)*. A total of 100 µL of the supernatant was mixed with 3.4 mL of the reaction mixture that consisted of 2.8 mL of $Na_2HPO_4$ and $NaH_2PO_4$ (0.05 mol/L pH 7.8) buffer and 100 µL of 0.1 mol/L of $H_2O_2$ solution and 0.5 mL of 2 mmol/L EDTA. The absorbance at 240 nm was recorded for 3 min and the attenuation of the absorbance was used to calculate the CAT activity against a calibration curve generated with $H_2O_2$. For the control group, 100 µL of 0.05 mol/L pH 7.8 phosphate buffer was used instead of the supernatant. Absorbance at 240 nm was recorded.

## POD activity

The peroxidase (POD) activity was determined as described by *Zhang (2003)*. A total of 3 mL of reaction solution contained 1 mL of 0.3% $H_2O_2$, 0.95 mL of 0.2% guaiacol, 1 mL of 50 mmol/L phosphate buffer (pH 7.0) and 0.05 mL enzyme extract, and the reaction was started with the addition of the enzyme extract. For the control group, 50 µL of 0.05 mol/L phosphate buffer (pH 7.8) was used instead of the crude enzyme extract. The changes in absorbance at 470 nm were recorded for 1 min.

## Determination of plant hormones

The concentrations of endogenous hormones, including IAA, ABA, GA, and zeatin (ZT) in leaves and roots were determined according to the methods previously reported by *Marasek-Ciolakowska et al. (2021)*. Frozen leaves or roots (2.5 g) were ground with a mortar and pestle to a powder in liquid nitrogen, then the powder was rapidly transferred to a 50 mL centrifuge tube and extracted with 20 mL of 80% methanol at 4 °C overnight. The extract was centrifuged at $12,000 \times g$ and 4 °C for 15 min. Supernatant was transferred into a 50 mL centrifuge tube and the residue was ultrasonically extracted with 15 mL of 80% methanol at room temperature for 30 min and centrifuged at $12,000 \times g$ and 4 °C for 15 min. Two supernatants were pooled, and concentrated to 20 mL in a rotary concentrator at 40 °C. Then, decolorization of the concentrate was carried out by adding and discarding two changes of 15 mL petroleum ether. The volume of the solution was further concentrated to near dry, and 2 mL of 80% methanol was then added and the solution mixed. Concentrations of endogenous hormones were determined using a HPLC-MS/MS (Agilent Infinity 1260, Agilent, Germany).

## Statistical analysis

There are four factors in the experimental design of this study: sampling year (2018 and 2019), plant age (3, 4, 5, 7, and 8), plant part (leaf and root), and physiological stage of plants (jointing and heading stage). Sampling year and plant age were confounded because the plant age differed between the 2 years, as well as climate conditions. We applied correlation analysis initially to the chemical and biochemical measures between leaves and roots and found there were significant correlations ($P < 0.05$) between two plant parts for the majority of the measures. There was also a significant correlation between the jointing and heading stages as leaves were collected from the same plants. These correlations were resulted from the biological associations between the leaves and roots, and between the jointing and heading stages from the same plants. Based on these outcomes, sampling year, plant part and physiological stage were not included as the factors in the statistical model, and the effect of plant age, which was the core focus of this study, on all the measures was examined, respectively, for each of the sampling years, plant parts, and physiological stages. The effects of plant age on all the measures were analyzed using the procedure of one-way analyses of variance (ANOVA) in SPSS (version 20.0; IBM, SPSS Inc., Chicago, IL, USA) for each of sampling years (2018 and 2019), plant parts (leaves and roots), and plant physiological stages (jointing and heading stages). Plant age was the fixed factor and the plot was a random factor. LSD multiple comparisons were undertaken to distinguish differences between the means. Data are present as the least square means and standard error of means (SEM). Statistical significance was declared with $P$ values < 0.05.

# RESULTS

## The aboveground biomass and seed yield

The aboveground biomass and seed yield for 3-, 4-, and 5-year old plants are shown in Table 3. The aboveground fresh and dry weights and seed yield declined substantially and continuously with plant ages: the fresh biomass of 4- and 5-year old plants declined by

**Table 3 Aboveground fresh and dry biomasses and seed yield of *Elymus sibiricus* at the anthesis stage.**

| Plant age | Fresh biomass (kg/ha) | Dry biomass (kg/ha) | Seed yield (kg/ha) |
|---|---|---|---|
| 3 | 15,632[a] ± 162.80 | 7,550[a] ± 163.28 | 809[a] ± 24.46 |
| 4 | 10,283[b] ± 171.03 | 5,823[b] ± 155.17 | 706[b] ± 40.54 |
| 5 | 7,434[c] ± 133.18 | 4,780[c] ± 162.88 | 532[c] ± 38.69 |

**Note:**
The data are presented as mean ± SE. Different superscripts within the same column indicate significant different between plant ages ($P < 0.05$).

34.2% and 52.5% respectively compared with 3-year old plants, and the seed yield declined by 12.7% and 34.2%, respectively.

## Relative water content (RWC), SPAD-chlorophyll and net photosynthesis (Pn)

The detailed data for the relative water content, SPAD-chlorophyll, and net photosynthesis (Pn) for 3-, 4-, and 5-year old plants in 2019 has been published elsewhere (*Jin et al., 2021*). In brief, RWC was 51.7%, 43.3%, and 35.6% respectively for 3-, 4-, and 5-year old plants ($P < 0.05$), the SPAD-chlorophyll value was 33.91, 37.27, and 32.83 ($P < 0.05$), and Pn was 7.73, 6.35, and 2.08 μmol/m$^2$·s. RWC and Pn declined significantly with the plant age ($P < 0.05$).

## Superoxide anion radical generation rate in leaves and roots

$O_2^{\cdot-}$ generation rate in leaves and roots is shown in Fig. 2. In 2018, $O_2^{\cdot-}$ generation rate in leaves of 7-year old *E. sibiricus* compared to 3- and 4-year old plants had declined by 50.5% and 55.0%, respectively at the jointing stage ($P < 0.05$), and by 53.8% in 3-year old plants at the heading stage ($P < 0.05$). In 2019, $O_2^{\cdot-}$ generation rate in leaves of 4-year old plants was declined by 8.9% and 9.7% respectively compared with those in 5- and 8-year old plants at the jointing stage ($P < 0.05$). At heading stage, $O_2^{\cdot-}$ generation rates in leaves of 5- and 8-year old plants were decreased by 38.9% and 27.4% respectively compared to those of 4-year old plants ($P < 0.05$) (Fig. 2A). In roots (Fig. 2B) harvested in 2018, $O_2^{\cdot-}$ generation declined continuously in plants aged from 3- to 7-years at the jointing state ($P < 0.05$). At heading stage, $O_2^{\cdot-}$ generation rates from 4- to 7-year old plants had declined by 54.2% and 4-year old plants were 3.3% less than 3-years (P < 0.05). In year 2019, $O_2^{\cdot-}$ generation rate declined with age in plants aged 4-, 5-, and 8-years at the jointing state ($P < 0.05$), but did not differ significantly at the heading stage between 5-year and 8-year old plants.

## MDA concentration in leaves and roots

MDA concentrations in leaves and roots are shown in Figs. 2C and 2D. In 2018 (Fig. 2C), the concentration in leaves was lower in 4-year old plants at the jointing stage than 3-year old plants ($P < 0.05$) and was further increased in 7-year old plants ($P < 0.05$). At the heading stage, the concentrations in 3- and 7-year old plants were 63.8% and 66.3% higher than that in 4-year old plants ($P < 0.05$). In year 2019, MDA concentration showed an increasing trend with plant age at both jointing and heading stages. For roots in 2018

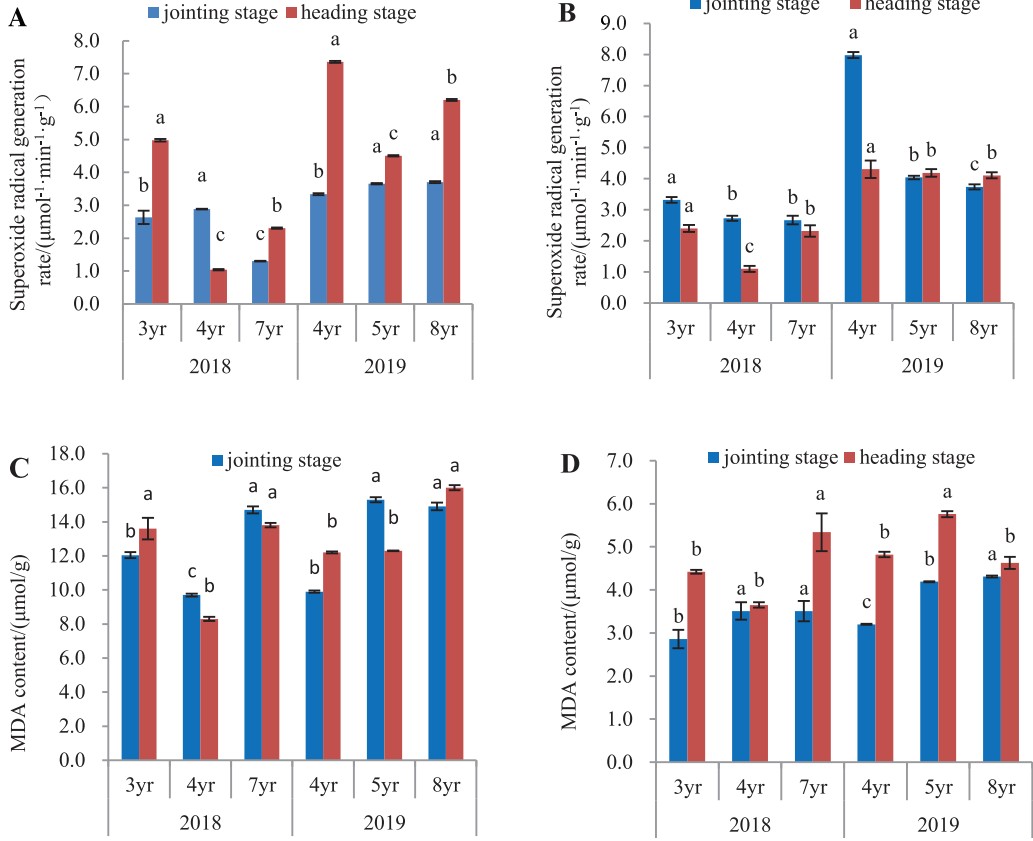

**Figure 2 Superoxide anion radical generation rate and MDA concentration in leaves and roots at the jointing and heading stages in 2018 and 2019 of *Elymus sibiricus*.** Vertical bars represent standard error (SE). Different letters for each parameter indicate significant differences between plant ages within the year ($P < 0.05$). (A) Superoxide anion radical generation rate in leaves; (B) superoxide anion radical generation rate in roots; (C) MDA concentration in leaves; (D) MDA concentration in roots.

(Fig. 2D), MDA concentrations in 4- and 7-year old plants were respectively 22.7% and 22.7% higher than that in 3-year old plants at the jointing stage ($P < 0.05$), and at the heading stage, the concentration in 7-year old plants was 20.8% and 46.3% higher than those in 3- and 4-year old plants ($P < 0.05$). In 2019, MDA concentration increased continuously with the increases of plant ages at the jointing stages. At the heading stage, MDA concentration in 5-year old plants was 19.5% and 24.4% higher than those in 4- and 8-year old plants ($P < 0.05$) respectively, but no difference in MDA concentration was found between 4- and 8-year old plants ($P > 0.05$).

## Antioxidant enzyme activities

The antioxidant activity of *E. sibiricus* was determined in both the leaves and roots, by assaying the activities of SOD, POD and catalase (Fig. 3).

## SOD activity

In 2018, SOD activity (Fig. 3A) in leaves of 7-year old plants was 30.2% and 28.9% higher than those of 3- and 4-year old plants, respectively at the jointing stage ($P < 0.05$), and 6.6%

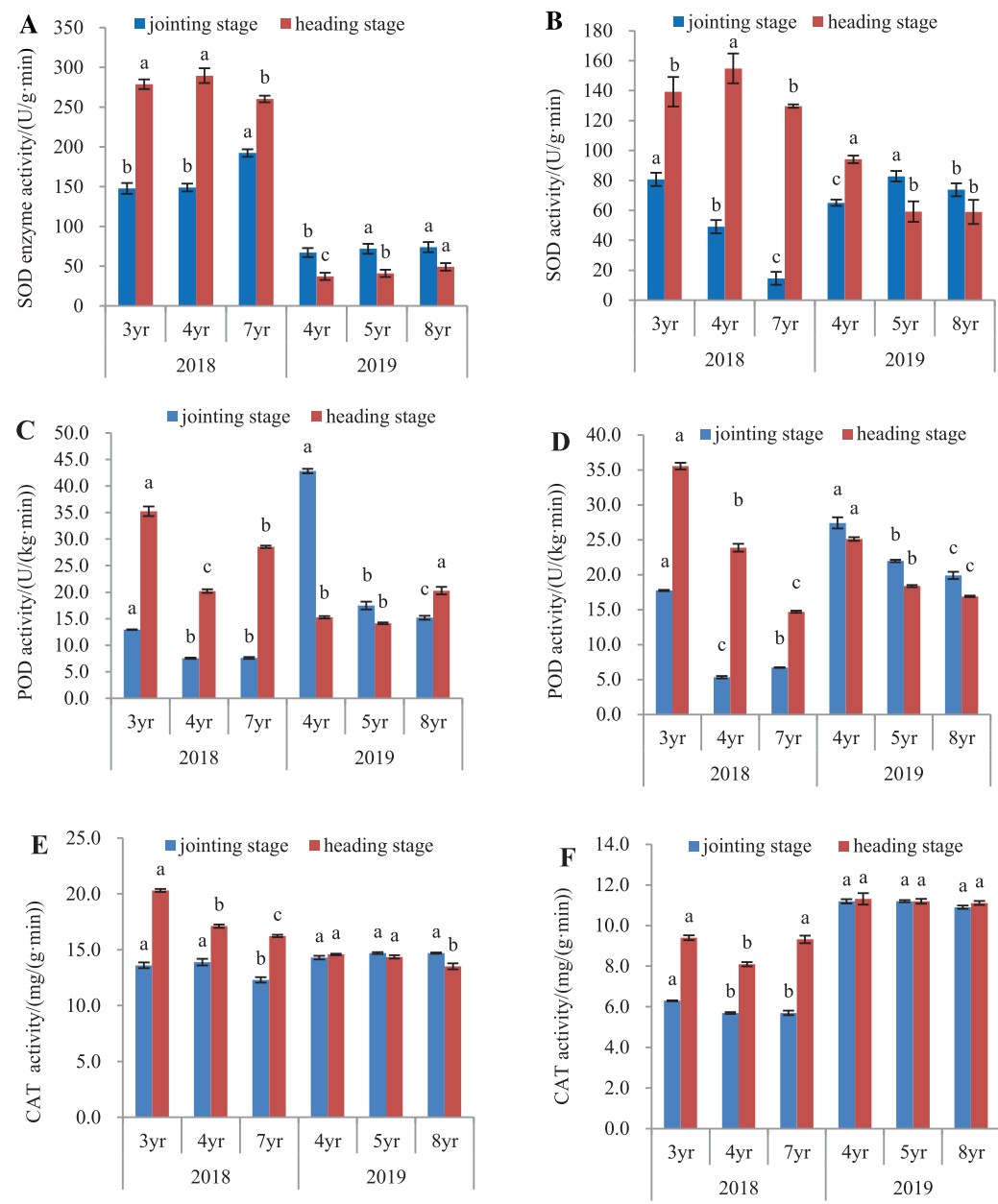

**Figure 3 SOD, POD and CAT activity in leaves and roots of *Elymus sibiricus* at the jointing and heading stages in 2018 and 2019.** Vertical bars represent standard error (SE). Different letters in each parameter indicate significant differences between plant ages within the year (*P* < 0.05). (A) SOD activity in leaves; (B) SOD activity in roots; (C) POD activity in leaves; (D) POD activity in roots; (E) CAT activity in leaves; (F) CAT activity in roots.

and 10.2% lower at the heading stage (*P* < 0.05). In 2019, SOD activity in leaves of 4-year old plants was 6.7% and 10.1% lower than those in 5- and 8-year old plants at the jointing stage (*P* < 0.05), and at the heading stage, the activity increased with plant age. For roots in 2018 (Fig. 3B), SOD activity declined continuously with the increases in plant ages at the jointing stage (*P* < 0.05). At the heading stage, root SOD activity in 4-year old plant was 11.2% and 19.4% higher than those in 3- and 7-year old plants (*P* < 0.05). In 2019, SOD

activity in 4-year old plants were 21.4% and 11.8% lower than measured in 5- and 8-year old plants, respectively at the jointing stage ($P < 0.05$), and at the heading stage, SOD activities in 5- and 8-year old plants were 37.1% and 37.3% lower, respectively than that in 4-year old plants ($P < 0.05$).

## POD activity

In 2018, POD activities in leaves of 4- and 7-year old plants at the jointing stage were 41.7% and 41.3% respectively, lower than that in 3-year old plants ($P < 0.05$), and at the heading stage, the activity in 4-year old plants was 42.6% and 29.2% lower than that in 3- and 7-year old plants, respectively ($P < 0.05$). In 2019, POD activity declined continuously with plant age at the jointing stage, but at the heading stage, the activity in 8-year old plants increased respectively by 32.7% and 43.4% when compared with those in 4- and 5-year old plants (Fig. 3C). For roots (Fig. 3D) in 2018, POD activity in 4- and 7-year old plants was respectively 69.9% and 62.1% lower than that in 3-year old plants at the jointing stage ($P < 0.05$), and at the heading stage, the POD activity declined continuously with the increase of plant age. In 2019, the POD activity declined continuously with the increased plant age at both the jointing and heading stages.

## CAT activity

The CAT activity in leaves and roots is shown in Fig. 3. In 2018, CAT activity in leaves of 7-year old plants was 9.6% and 11.5% lower than those in 3- and 4-year old plants, respectively at the jointing stage ($P < 0.05$). At the heading stage, the activity declined with the increase of plant age. In 2019, however, no difference was found in the CAT activity in plants aged of 4 and 5 years at both the jointing and heading stages ($P > 0.05$), except for the 7.27% and 5.85% lower CAT activity in 8-year old plants compared to 4- and 5-year old plants, respectively at the heading stage (Fig. 3E). For roots in 2018 (Fig. 3F), CAT activity in 3-year old plants was 9.5% lower than those in 4- and 7-year old plants at the jointing stage ($P < 0.05$), and at the heading stage, the CAT activity in 4-year old plants was 13.8% and 13.1% lower than those in 3-year and 7-year old plants, respectively. In 2019, no difference was found in the CAT activity in plants aged of 4, 5, and 8 years at both the jointing and heading stages ($P > 0.05$).

## Endogenous hormones in leaves and roots

The endogenous hormones of *E. sibiricus* were determined in both the leaves and roots, by assaying the concentrations of IAA, ZT, GA and ABA (Fig. 4).

## IAA concentration

The IAA concentrations in leaves and roots is shown in Figs. 4A and 4B. In 2018, IAA concentration in leaves increased continuously with the increase of the plant age at both the jointing and heading stages. In 2019, this increasing trend of concentration with plant age was also present in the leaves of 4-, 5-, and 8-year old plants at the jointing stage, as well as in 4- and 5-year old plants at the heading stage. The IAA concentration in 8-year old plants dropped by 95.7% and 97.5% compared with those in 4- and 5-year old plants, respectively ($P < 0.05$) (Fig. 4A). For roots in 2018, IAA concentration in 3-year old plants

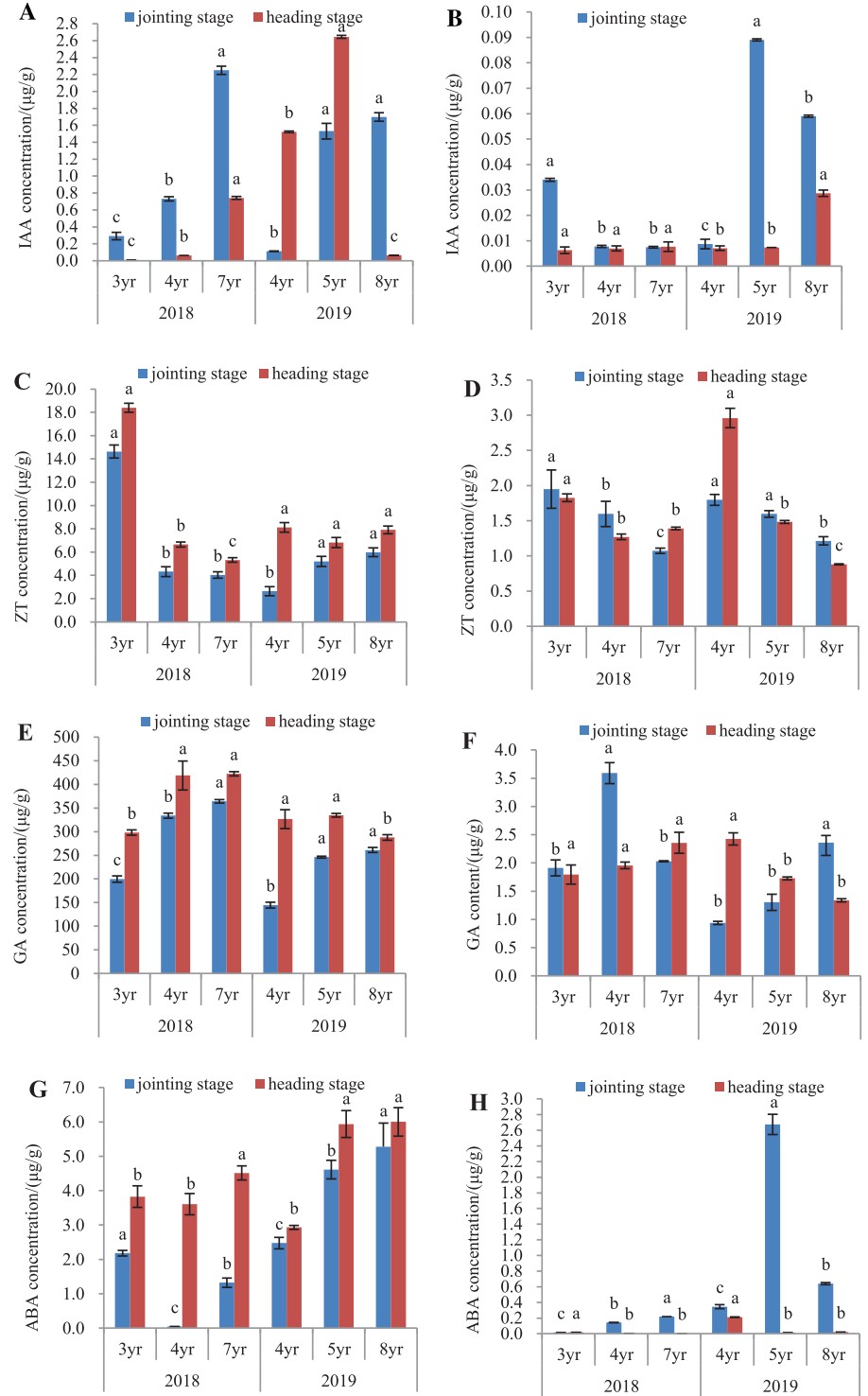

**Figure 4 IAA, ZT, GA and ABA concentrations in leaves and roots of *Elymus sibiricus* at the jointing and heading stages in 2018 and 2019.** Vertical bars represent standard error (SE). Different letters in each parameter indicate significant differences between plant ages within the year ($P < 0.05$). (A) IAA concentration in leaves; (B) IAA concentration in roots; (C) ZT concentration in leaves; (D) ZT concentration in roots; (E) GA concentration in leaves; (F) GA concentration in roots;(G) ABA concentration in leaves; (H) ABA concentration in roots.       

was 3.25-fold of the concentrations in 4- and 7-year old plants at the jointing stage ($P < 0.05$), and at the heading stage, there was no difference in the IAA concentration between plant ages ($P > 0.05$). In 2019, IAA concentrations in 5- and 8-year old plants were 8.89- and 5.56-fold, respectively of that of 4-year old plants at the jointing stage ($P < 0.05$). At the heading stage, the IAA concentration in 8-year old plants was 3.14-fold of those in 4- and 5-year old plants ($P < 0.05$) (Fig. 4B).

### ZT concentration

In 2018, the ZT concentrations in leaves of 3-year old plants were 2.39- and 2.62-fold higher of those in 4- and 7-year old plants, respectively at the jointing stage ($P < 0.05$), and at the heading stage, the concentration declined continuously with increasing plant age. In 2019, ZT concentrations in leaves of 5- and 8-year old plants were 97.3% and 126.5%, respectively higher than that of 4-year old plants at the jointing stage ($P < 0.05$). At the heading stage, there was no significant difference in the ZT concentration with plant age ($P > 0.05$) (Fig. 4C). For roots in 2018 (Fig. 4D), ZT concentration declined continuously with the increase of plant age at the jointing stage, and at the heading stage, the concentrations in 4- and 7-year old plants declined by 30.6% and 24.7%, respectively compared with that in 3-year old plants ($P < 0.05$). In 2019, the concentration showed a decline trend with plant ages at both the jointing and heading stages.

### GA concentration

The GA concentration in leaves and roots is shown in Figs. 4E and 4F. In 2018 (Fig. 4E), GA concentration in leaves showed an increase with the increase of plant age at the jointing stage, and also at the heading stage, but there was no significant difference in the concentration between 4- and 7-year old plants. In 2019, GA concentrations in leaves of 5- and 8-year old plants were 70.0% and 80.3%, respectively higher than that in 4-year old plants at the jointing stage ($P < 0.05$). At the heading stage, the concentration in 8-year old plants was 11.9% and 13.96% lower than those in 4- and 5-year old plants (Fig. 4E). For roots in 2018, GA concentration in 4-year old plants was 88.0% and 76.8% higher than those in 3- and 7-year old plants, respectively at the jointing stage ($P < 0.05$), and at the heading stages, there was no significant difference in the GA concentration with plant age ($P > 0.05$). In 2019, GA concentration showed an increasing trend with plant age at the joint stage, but at the heading stage, there was a declining trend with plant age (Fig. 4F).

### ABA concentration

The ABA concentration in leaves and roots is shown in Figs. 4G and 4H. In 2018 (Fig. 4G), the ABA concentration in leaves was the lowest in 4-year old plants at the jointing stage, followed with that in 7-year old plants, and 3-year old plants had the highest concentration. At the heading stage, ABA concentration in leaves of 7-year old plants was 18.0% and 25.2% higher than those in 3- and 4-year old plants, respectively ($P < 0.05$). In 2019, the ABA concentration showed an increase with plant age at both the jointing and heading stages. The ABA concentrations in roots (Fig. 4H) were much lower compared with those in leaves. For roots in 2018, ABA concentration showed an increase with plant

**Table 4 The effects (F values and statistical significances) of plant age and its interactions with plant physiological stage on antioxidant indices and plant hormones analyzed in two-way ANOVA model.**

| | 2018 | | | 2019 | | |
|---|---|---|---|---|---|---|
| | Plant age | Physiological stage | Age × Stage | Plant age | Physiological stages | Age × Stages |
| **Leaves** | | | | | | |
| $O_2^{\bullet-}$ | 6,215*** | 8,933*** | 7,051*** | 1,254*** | 13,405*** | 1,872*** |
| MDA | 172.1*** | 0.794[ns] | 13.38*** | 566.8*** | 1.572[ns] | 217.0*** |
| SOD | 7.853** | 1,588*** | 69.732*** | 45.41*** | 1,279*** | 5.786* |
| POD | 312.1** | 3,040** | 82.72** | 418.3** | 452.4** | 586.7** |
| CAT | 74.13*** | 742.0*** | 37.63*** | 4.231* | 11.16** | 11.62** |
| IAA | 1,155*** | 1,138*** | 221.6*** | 541.2*** | 68.71*** | 744.8*** |
| ABA | 19.76*** | 269.2*** | 11.85** | 34.38*** | 7.000** | 0.659[ns] |
| ZT | 659.2*** | 68.463*** | 5.831* | 7.782** | 85.54*** | 14.59*** |
| GA | 69.04*** | 53.75*** | 1.144[ns] | 17.95*** | 166.3*** | 34.56*** |
| **Roots** | | | | | | |
| $O_2^{\bullet-}$ | 2,010*** | 6,232*** | 909.5*** | 4,413*** | 2,418*** | 3,772*** |
| MDA | 1.329** | 6.254*** | 1.243** | 94.26*** | 417.6*** | 55.58*** |
| SOD | 12.23*** | 196.5*** | 7.004** | 32.08*** | 5.183* | 142.4*** |
| POD | 1,428*** | 3,383*** | 178.4*** | 203.7*** | 81.87*** | 1.371* |
| CAT | 35.15*** | 1,251*** | 18.60*** | 1.77[ns] | 0.524[ns] | 0.212[ns] |
| IAA | 77.91*** | 91.97*** | 87.92*** | 1,064*** | 737.2*** | 1,799*** |
| ABA | 3,396*** | 15,089*** | 4506*** | 249.0*** | 668.8*** | 309.8*** |
| ZT | 12.06*** | 0.158[ns] | 2.883[ns] | 175.0*** | 16.52** | 63.04[ns] |
| GA | 12.23*** | 9.564** | 14.85*** | 1.542[ns] | 3.664[ns] | 22.08*** |

**Notes:**
* $P < 0.05$.
** $P \leq 0.01$.
*** $P \leq 0.001$.
ns not significant.

age at the jointing stage, and at the heading stage, the concentration was almost negligible. In 2019, ABA concentration was much higher in 5-year old plants, followed with that of 8-year old plants, and 4-year old plants had the lowest ABA concentration at the jointing stage. At the heading stage, ABA concentration in 4-year old plants was 13.1-fold and 9.1-fold of the concentrations of 5- and 7-year old plants, respectively ($P < 0.05$).

## Interactions between plant age and physiological stages on antioxidant indices and plant hormones

We measured the antioxidant indices and plant hormones in leaves and roots of different ages *E. sibiricus* at the jointing and heading stages in this study. To explore possible interactions between the plant age and physiological stages on these measures, two-way ANOVA model was also used to re-analyze each of the variables within 2018 and 2019, respectively. The results (F values and their corresponding statistical significances) are shown in Table 4. For the leaves, plant age and its interaction with the physiological stage

had significant effects on $O_2^{\bullet-}$, MDA, SOD, POD, CAT, IAA, ABA, ZT, and GA ($P < 0.05$), except for the interactions on GA in 2018 and ABA in 2019 ($P > 0.05$). As for roots, plant age had significant effects on $O_2^{\bullet-}$, MDA, SOD, POD, CAT, IAA, ABA, ZT, and GA in 2018 ($P < 0.05$), as well as these parameters in 2019 except for CAT and GA in 2019 ($P > 0.05$). The interactions between plant age and the physiological stage on all these parameters were significant ($P > 0.05$) except for the effects on ZT in 2018, and CAT and ZT in 2019 ($P > 0.05$). Since there were significant correlations between the two physiological stages on the majority of these parameters measured (results are not reported), the effects of the physiological stages are not described in this article.

## DISCUSSION

Rapid deterioration of vegetation status and declines of aboveground biomass with the age of *E. sibiricus* plants were observed in this study, indicating that *E. sibiricus* aged at a high rate, which has been commonly found in other reports (*Jin et al., 2021*; *Yang et al., 2021*). Plant production and seed reproduction capacities decrease with increase of plant age, which affects the maintenance and regeneration of the plant population (*Kuai, 2014*).

Likely, increase of plant senescence contributes to these declines. Senescence occurs at different stages and at different plant components (plant structure, organs, tissues, cells, *etc.*) (*Leopold, 1961*). In the process of senescence, a series of changes occur in the external morphological characteristics of each plant part (*Van Doorn & Woltering, 2004*). These changes mainly include plant height, leaf number and biomass, and the changes of these morphological features on the surface ultimately reflect the changes of physiological and biochemical processes and material transport inside the plant that will eventually affect the yield and quality of the seed (*Kuai, 2014*; *Song, 1998*).

The present study showed that the RWC and photosynthetic rates in leaves of *E. sibiricus* reduced remarkably with the increase of plant age. Plants will stop growing when their ability to transport water to their leaves becomes insufficient, as reflected by the reduction of RWC. Studies from a variety of experimental situations and plant species indicate that stomatal conductance and consequently photosynthetic rates are reduced with plant age (*Munné-Bosch & Alegre, 2002*; *Kolb & Stone, 2000*). Thus, *Munné-Bosch & Alegre (2002)* proposed that the reductions in photosynthesis could lead to an enhanced oxidative stress associated with the plant aging process.

Plant aging is one of the most crucial and complex physiological phenomena in the lifecycle of a plant, and plants often fall prey to environmental and biological stresses that can lead to erratic growth. An increase in oxidative stress is one of biological stress indicators that are linked to plant aging (*Munné-Bosch & Alegre, 2002*). Oxidative stress can occur when the rate of scavenging free radicals is over-ridden by the rate of free radical production in an organism. In the present study, we measured the superoxide anion radical generation rate in leaves and roots at the jointing and heading stages, but did not find any apparent pattern between the superoxide anion radical generation rate and plant age. The radical generation rates in both leaves and roots in the heading stage were greater than those in the jointing stage. There was also a large year-to-year variation in the superoxide anion radical generation rate (Figs. 2A and 2B). It should be noted that the
year-to-year variation confounds the effects of plant age by environmental changes (climates in particular) between years. Therefore, the results suggest that the superoxide anion radical generation rate was influenced strongly by environmental factors as well as the physiological stage of plants (*Noctor & Foyer, 2016*; *Qin et al., 2018*).

The concentration of MDA is an indicator to oxidative damage of lipids in plants (*Ozlem, 2022*). The results in the present study showed an increasing trend in the MDA concentration with plant age, particularly in leaves and roots at the heading stage in 2019, indicating an increase of lipid peroxidation with plant age (Figs. 2C and 2D). With increasing of plant age, the activities of SOD, POD and CAT of antioxidant system decreased, implying that the ability of scavenging reactive oxygen species was weakened, thus the degree of membrane lipid peroxidation was deepened and the MDA concentration increased (*Wang et al., 2013*). An increase of MDA concentration is a result of damage to the membranes and accelerated aging, leading to the metabolic dysfunction of plant cells and can even lead directly to cell death (*Rysz et al., 2022*). Interestingly, it was noted that the physiological state had a significant influence on the MDA concentration in roots but not in leaves (Table 4); also the year-to-year variation in the MDA concentration appeared to be small. It seems that the MDA concentration was associated with the age of roots, as well as the physiological stage.

The antioxidative defense system comprises of several antioxidant enzymes such as SOD, catalase, and POD that scavenge superoxide anion radicals, peroxides, and other free radicals in plants (*Noctor & Foyer, 1998*). The results in this study showed that the SOD activity appeared to decline with plant age in roots, but not in leaves (Figs. 3A and 3B); the POD activity declined with plant age in both leaves and roots (Figs. 3C and 3D); whereas the CAT activity declined with plant age in leaves at the heading stage in 2018, but remained almost unchanged with plant age in both leaves and roots at the other plant stages and years (Figs. 3E and 3F). Overall, the antioxidant capacity appeared becoming weak with plant aging, particularly in roots. As noted, the superoxide anion radical generation rate did not change significantly with plant aging, so the weakness of the antioxidant capacity could result in a risk of oxidative stress, and concurs with the increasing MDA concentration with plant age in this study. These results indicate that the overall oxidative capacity was affected by plant aging, in agreement with previous studies (*Munné-Bosch & Lalueza, 2007*). It is also possible that oxidative stress accelerated leaf senescence with plant aging, therefore, is regarded as an adaptive strategy for plants to cope with environmental stresses (*Munné-Bosch, Jubany-Marí & Alegre, 2001*).

It has been reported that plant endogenous hormones are one of the important factors that regulate plant senescence (*Jan et al., 2019*). However, little is known about their roles in plant aging process. In this study, we determined the concentrations of IAA, GA, ZT, and ABA in both leaves and roots at the jointing and heading stages. Overall, the concentrations of these hormones were significant lower in roots than in leaves, particularly IAA, GA, and ABA (Fig. 4). IAA is involved in the regulation of leaf expansion and newly emerged leaves to produce auxins (*Aldés, Centeno & Fernández, 2004*). ZT is a type of cytokinin participating in many physio-biochemical processes, including different cellular divisions and the senescence of leaves, thus regulates the ratio of shoot/root

systems (*Azzam et al., 2022*). A reduction of such cytokinins is associated with plant aging in conifers (*Valdés, Fernández & Centeno, 2003*). GA is necessary for shoot and root elongation and generally associated with plant senescence (*Ptošková et al., 2022*). ABA regulates various developmental processes and serves as an inducer to trigger plant senescence (*Lim, Kim & Nam, 2007*; *Asad et al., 2019*). In the present study, the IAA concentration presented different patterns with plants age between leaves and roots. In leaves, the IAA concentration increased with plant age, except for the very low concentration in 8-year old plants, the most of which was dead plant matter; whereas in roots, there was no clear pattern between the IAA concentration with plant age (Figs. 4A and 4B). The ZT concentration in roots at both the jointing and heading stages declined with plant age (Figs. 4C and 4D). The changes of the GA concentration with plant age varied between the physiological stages and between years: increasing with plant age in leaves and roots at the jointing stage in 2018, otherwise, no clear pattern was seen. The GA concentration is usually low in roots (Figs. 4E and 4F), however, such a low concentration can maintain root growth (*Ptošková et al., 2022*). A reduction of GA could decrease the capacity of growth as plants age (*Colebrook et al., 2014*). Previous studies have shown that the ABA concentration in plants was determined by the balance between ABA biosynthesis and catabolism (*Zhang et al., 2018*). It is known that roots are an important site of ABA synthesis, and then ABA is transported from roots to leaves through the xylem vessel (*Wilkinson & Davies, 2002*; *Dodd, 2005*). The increase of endogenous ABA can reduce transpiration of the plant by inducing stomatal closure, but also by decreasing leaf area. The ABA concentrations in leaves and roots appears to be increasing with plant age particularly in leaves, albeit the data for ABA in roots for 2019 is unclear (Figs. 4G and 4H).

Aging occurs throughout the lifetime of perennials at the tissue and organ levels (*Jing, Hille & Dijkwel, 2003*) in both above- and underground parts. Stems and leaves are mainly annually produced, while roots are mainly perennial with ongoing new growth in a perennial plant. *E. sibiricus* is a typical perennial species. The differences in physiological characteristic of roots reflect not only the changes within the growing season, but also the response to the growing years, and their living conditions will affect the growth of plants in the next season (*Wang, 2014*). Leaf senescence has been studied intensively. However, information about the mechanisms of roots in plant aging is not so well understood yet. It was proposed that root senescence is closely related to leaves senescence, and the main reasons provided are that root-tips are the site of synthesize cytokinins and gibberellins, which are then transported upward through the stem and leaf to regulate the senescence of stems and leaves. As the vitality of roots decreases, so does the ability to synthesize hormones, resulting in a decline in the anti-aging ability of the aboveground part, leading to aging (*Chen & Brassard, 2013*). Based on the literature, we believe physiological and biochemical changes in roots may play a primary role in the plant aging process. Aforementioned, the age-associated reduction of antioxidant capacity, particularly the SOD and POD activities, in roots could be one of the contributors; the decline of the ZT concentration and a tendency of increasing ABA concentration in roots cannot be ruled out.

Aging processes in plants are highly complex physiological processes that are influenced directly or indirectly by extrinsic factors and intrinsic changes during plant development. The process becomes more complex due to the responses of perennials to climate variations from year to year. However, it has become clear that some meristem-associated pathways can integrate information from small peptides, enzymes, hormones, age-related signaling, and environmental cues in plant development (*Bustamante, Matus & Riechmann, 2016*; Li, Qi & Yang, 2022). These findings highlight the effects of the plant age, growth stage, and their interactions on the antioxidant activity and endogenous hormones of *E. sibiricus* (Table 4). Future studies may explore more underlying age-related changes in *E. sibiricus* at different locations or environments.

# CONCLUSION

*E. sibiricus* grasses showed a rapid aging process with substantial reductions of aboveground biomass and seed yield with plant age. The aging process appeared to be associated with the reduced activities of SOD and POD in roots and the increase of oxidative stress as indicated by increased MDA concentration in roots and leaves. The plant hormone concentrations were many-fold lower in roots than those of the leaves. Among the root hormones, ZT concentration appeared to increase while ABA concentration tends to decline with plant age. However, these plant age-related trends were influenced significantly by plant physiological stages and year-to-year variations, likely due to climate differences between years. In future studies, long-term field experiments need to be undertaken out to distinguish and minimize these influences to explore the regulatory mechanisms in the aging process of *E. sibiricus*.

## Funding

This work was supported by the National Natural Science Foundation of China (grant number 31660684) and the Assessment of Carbon Storage and Carbon Sink Value of Artificial Grassland in Hexi Irrigation Area of Gansu Province (kjcx2022009). The funders had no role in study design, data collection and analysis, decision to publish, or preparation of the manuscript.

## Grant Disclosures

The following grant information was disclosed by the authors:
National Natural Science Foundation of China: 31660684.
Assessment of Carbon Storage and Carbon Sink Value of Artificial Grassland in Hexi Irrigation Area of Gansu Province: kjcx2022009.

## Competing Interests

The authors declare that they have no competing interests.

## Author Contributions

- Juan Qi conceived and designed the experiments, performed the experiments, analyzed the data, prepared figures and/or tables, authored or reviewed drafts of the article, and approved the final draft.
- Zhaolin Wu conceived and designed the experiments, performed the experiments, analyzed the data, prepared figures and/or tables, authored or reviewed drafts of the article, and approved the final draft.
- Yanjun Liu performed the experiments, analyzed the data, prepared figures and/or tables, and approved the final draft.
- Xiangjun Meng conceived and designed the experiments, authored or reviewed drafts of the article, and approved the final draft.

## Data Availability

raw data can be found in the Supplemental Files

## Supplemental Information

Supplemental information for this article can be found online at http://dx.doi.org/10.7717/peerj.15150#supplemental-information.

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
