# Peer review of "Effects of plant age on antioxidant activity and endogenous hormones in Alpine Elymus sibiricus of the Tibetan Plateau"

_PeerJ, doi:10.7717/peerj.15150_

## Round 0.1 · original submission · Major Revisions

Please, carefully read the reviewers' suggestions and check your article according to them. If any advice is unacceptable for your article, you should write why you do not accept it. The article needs proper language editing. So, you can use the editing service of PeerJ or take help from a colleague who is a fluent speaker and expert on these topics.

Reviewer 1 ·

Basic reporting

Thank you for choosing me as your reviewer. However, the current study is not suitable for publication in the journal due to its sloppy writing and lack of proving elements, although it covers an important issue.

Experimental design

The authors designed an experiment on an important study topic, but the experimental design was flawed.

Validity of the findings

Unfortunately, the subject of study is not well proven. The findings obtained do not prove the subject of the study. Indeed, antioxidative enzymes such as sod and cat are parameters that measure instant stress. As plants are not under controlled conditions, it may be wrong to associate changes in these enzymes with plant age. they could verify the work by analysis of non-enzymatic antioxidants.

·

Basic reporting

GENERAL COMMENTS

The aim of this study was to examine the effects of plant age on antioxidant activity and endogenous hormones in Elymus sibiricus in the Alpine region of the Tibetan Plateau. The language of this study is understandable and grammatically quite good. Therefore, the manuscript does not need to be edited in terms of language. Minor grammatical arrangements of the study were given in the text. The abstract, introduction, material methods, and discussion parts of the manuscript are written very well. Paragraph transitions are very convenient. However, the experimental design used is incorrect. Statistical analysis needs to be done again. After the analyses are renewed, the results and conclusion parts of the article should be revised again. Also, the references of the study were checked again, and the suitability of the journal format was left to the authors.

Experimental design

There are 4 factors in the experimental design of the study. These are year (2018-2019), plant age (3-4-5-6-7-8), plant parts (leaf-root), and sampling time (jointing-heading). It is understood that you do not use the factor of year and sampling time in the statistical model because you think that environmental factors have an effect. However, why did you not include the plant parts factor in the model? In this way, you would have investigated both the averages of plant age and plant component levels and the existence of plant age x plant parts interaction. If leaves and roots are taken from the same plants, the plant component factor is dependent, and you should arrange your experiment plan as "two factors experiments with repeated measurement on one-factor levels". If leaves and roots are not taken from the same plants, the plant component factor is independent and you should arrange your experiment plan as "Two-way ANOVA"

Validity of the findings

Since the statistical model has changed, you need to rewrite the results section.

Additional comments

I believe that the quality of the study will increase after all the corrections are made.

Reviewer 3 ·

Basic reporting

1. Basic Reporting
1.1 Authors must send the article for English editing, to improve the readability and ease of understanding the research.
1.2 The abstract needs revising, the problem is stated, but the significance/importance/need of the study is not elaborated under the abstract. See revised sentence under general comments.
1.3 Numbers describing percentage increases in traits (superoxide, MDA, enzymes and hormones) of the most important observations (highlights) must be included under the abstract.
1.4 The introduction is well written and includes sufficient literature to back up the study.
1.5 Figures are of high quality, but the figure legend for all figures is not clear, it requires more elaboration. Figures must also be merged to reduce the number of figures in the manuscript.

Experimental design

2. Experimental design
The experiments are well designed and seem easy to follow.

Validity of the findings

3. Validity
3.1 I cannot comment on the novelty of the study, since this is not my field, but the research has a national impact since the grass has benefits for livestock farming. Thus, understanding factors that lead to early aging is important for the continuous cultivation of this grass.
3.2 The manuscript lacks some experimental data that will support the findings of current study, for example analysis of relative water content and the photosynthetic rate or photosynthetic pigments to support the data obtained under biomass. Osmolyte content analysis of either proline or soluble sugars can shed some light into the water related or osmoregulation capacity of the plants.
3.3 The results provided in the manuscript are of good quality, well interpreted and statistically analyzed.

Additional comments

4. General comments
Abstract:
● The importance of Elymus sibiricus must be clearly described.
● See added sentence on line 13“Elymus sibiricus L., is a perennial herb that has potential to serve as a grassland for livestock farming”. Elymus sibiricus shows rapid and substantial reductions of aboveground biomass and seed yield after 3 or 4 years of growth, and has an accelerated aging process.
● Line 18-19, sounds as if authors only assayed the antioxidant enzymes and hormones, sentence can be revised “Determining the influence of oxidative damage and hormones signa ling”.
● Appeared? Should be deleted.
● What is the meaning of many-fold? Give a number e.g if you want to generalize rather give a value that is close e.g 50-fold, 10-fold lower etc..
● Under the results section, from line 19 to 31 authors should consider revising this section and include % increase or decrease rather than only stating there was a decrease or increase.

Introduction:
Generally, the introduction is good , except that it requires revision by an English editor.
● Line 44: needs revising, by an English editor
● Line 54 grammar
● Line 60, ROS should be written in full as is mentioned for the first time.
● Line 68, abbreviation for superoxide anion is incorrect
● Line 72, spelling error
● Line 74 – 75, incomplete sentence
● Line 76-80, needs revising as it is not clear.


Methods:
● Experiments missing: in addition to the relative water content/photosynthesis/ osmolyte content analysis, element analysis experiments of how nutrients are absorbed and distributed, can also shed light into aging of this crop.
● By understanding these parameters, it will now be easy to predict future research, which can lead to the restoration of these metabolic processes.
● Clearly indicate where the E. sibiricus seeds were obtained and include other seeds information.
● Line 126 “Leaves of similar parts were collected from the first to third leaves (counting from the tip of each branch) on each of the branches”. Should be changed to leaves showing similar traits were
● Line 131 should be revised, “in” should be replaced with “during”
● Overall, lines 131 to 135, should be revised to sound much better.
● Under sample processing and assays there should be subheadings same as line 177
o Determination of lipid peroxidation for MDA content
o Determination of antioxidants enzymes
SOD activity
CAT activity
POD activity
Results:
● Line 212 has spelling error, please revise
● Revise the sentence: Line 217, “The concentration for 5-year-old plants was lower than that for 4- and 8-year old plants at the heading stage”
● Under O2- generation rate in leaves and roots, and following sections authors should include % for the obtained results as done on the above sections.
● Revise line 268-269
● Figure 2: “Plant phenotypes of 3- (A) and 4-year old plant (B) and roots of 4-year old plants (C)”.
● This figure legend is not very clear, it needs to be revised.
● Include arrows to clearly show the “rotten” phenomenon that is highlighted in line 202. If authors could also measure the extent of rotten expansion highlighted in line 203-205 across all plant age investigated in this study, this will provide more quantitative data, which can be correlated with other traits.
● Indicate the month and year when the pictures were taken.
● The 4-year old plant is shown as a whole plant, showing the shoots and roots. There must be consistency in the representation of results, meaning the whole plant for the 3-year old must be also be shown not only the shoot.
● I understand that the pictures for other plants aged 5-8-years are lost but it will make scientific sense if (in Line 205 – 206) the phenotype of all plants (3, 4,5,7 & 8 years old) are included.
General results
● Line 212, Check Spelling of superoxide anion radical
● There is a lack of explanation and consistency as to why authors only have data for 3, 4 and 7-years old plants in 2018 and 4, 5- and 8-years old plants in 2019. In the beginning of each section, there must be a clear explanation of what to be expected, looking at the section describing the plant phenotype it needs to be explained that only plants of a particular age will be described and why? Same format must apply to other sections as well.
● It makes it so difficult to understand the mechanisms behind aging in these plants if some results are missing. I suggest that the focus must be on plants whose data is available for both years 2018 and 2019. Again, what about age 6 plants?
● What are the reasons for the 4 year plant having low superoxide in 2018, than the plant in 2019? Explain.
● My analysis of Figure 3: Younger plants are generating higher ROS than older plants e.g 3-year in 2018 and 4-year in 2019. Why? Is this what the authors are expecting?
● Lipid peroxidation is observed at older plants 7-year (2018) and 5-year and 8- year (2019) in the leaves, there is a little bit of correlation. Under discussion, authors should provide more clear scientific explanations rather than the influence of environmental factors.
● Line 240: Change subheading to: Antioxidant enzymatic activity of Elymus sibiricus
● Add this statement “The antioxidant activity of Elymus sibiricus was determined in both the leaves and roots, by assaying the activities of SOD, POD and catalase (Figure)”.
● The article has too many figures, combine the following, Figure 3: Superoxide and MDA contents, this is because they both represent oxidative damage, the subheading can be changed to Markers of oxidative damage or Oxidative stress markers
● All enzymes; SOD, POD and CAT on a single figure as SOD (A&B), POD (C&D), CAT (E&F), Figure 4.
● Since there is a decline in the antioxidant enzyme activity with an increase with plant age, lipid peroxidation results also showed that MDA contents increased with increasing plant age. This is an interesting findings, I therefore suggest that a statistical correlation analysis must be conducted for all the studied traits to fully understand this aging.
● Combine the following figures for the hormones: IAA (A&B) and ABA (C&D); ZT(A&B) and GA (C&D).

Discussion
● The discussion needs to be revised, it must link clearly to the results, e.g The results in this study showed that the SOD activity appeared (Figure),
● Line 398, what do you mean by many fold? and consistency is important under the biomass, authors used percentage increase, it should remain the same.
● Authors must interpret the results properly Line 300-304 must be revised.
● Based on the content of MDA it is clear that the plants were under stress, this is also corroborated by a high increase in antioxidant enzyme activities (check the normal activity of antioxidant enzymes in different ages of plants). This section must be thoroughly discussed.
● For Table 4: instead of having the p values, include either a correlation table or correlation figure, it is an easy way to show the relationship between the studied traits in explaining the hypothesis.

---

## Round 0.2 · Minor Revisions

Many thanks for your positive attitude towards the comments of reviewers. Please complete the revision and submit it to the system.

·

Basic reporting

As the only statistical point of criticism, "The Pearson correlation between leaves and roots at different growth stages in 2018 and 2019 was found to be statistically significant." this is expression. Since the Pearson correlation is statistically significant, it cannot be excluded from the statistical model. Nevertheless, I congratulate the authors for all the editing and development of their manuscript.

Acceptance is appropriate as it provides a sufficient improvement in the statistical analysis of the manuscript and, accordingly, in the organization of comments.

Experimental design

As the only statistical point of criticism, "The Pearson correlation between leaves and roots at different growth stages in 2018 and 2019 was found to be statistically significant." this is expression. Since the Pearson correlation is statistically significant, it cannot be excluded from the statistical model.

Validity of the findings

The progress in this section is sufficient.

Additional comments

I believe that this study will contribute to the literature in this way.

Reviewer 3 ·

Basic reporting

To authors:
I commend the authors for extensively attending to the reviewer’s comments, Authors have done exceptional fieldwork, which was carried over a period of 8 years from 2012 – 2019. The manuscript is now clearly written in professional and unambiguous language. If there is a weakness, it is minor grammatical errors (as I have noted in rebuttal-revised letter) which should be improved before acceptance for publication.

Experimental design

Experimental design has been improved.

Methods section Line 135-147 to be described in detail, and references must be provided especially for the relative water content.

Validity of the findings

This section is also well revised as indicated under the rebuttal letter. Some of my comments are also included there.

Additional comments

Once again, I commend the authors for extensively attending to the reviewer’s comments. This is exceptional fieldwork, which was carried over a period of 8 years from 2012 – 2019. This is an important study for world agriculture, thus securing food for livestock is also a priority, this kind of research must be encouraged and shared with a wide scientific community.

See my comments below based on the author's rebuttal letter.:

1. 1st review: Authors must send the article for English editing, to improve the readability and ease of understanding the research.
Author: Thanks very much for your comments. The manuscript has been edited for English language by a retired Principle Senior Research Scientist who is a native English speaker and used to work in biological areas for Commonwealth Scientific and Industrial Research Organisation (CSIRO).

2nd Reviewer reply: Thank you to the authors, the language has been improved, however there are still some few sentences that requires re-editing as I have indicated below, but I suggest that the entire document be revised fir grammatical errors.

Here are few sentences to be revised:

Line 26, abstract minor gramma, [English/language editing]
Line 27 Im still a little bit confused here, If there is 7 years old plants, what about the SOD data for 5 years
Line 57-59 “It is hypothesized that with increasing age E. sibiricus plants there will be intrinsic changes in physiological and biochemical metabolism that will affect the
plant physiology leading to a productivity reduction”.
Line 100-101, Research has shown that these
100 phytohormones do not work alone, and they are often functioning concomitantly to regulation plant senescence
Revised sentence: Research has shown that these phytohormones do not work alone, but and they are often functioning concomitantly to regulate ion plant senescence
Line 101-105, requires revision.
Line 141-147 in the PDF copy, a reference must be provided for the measurement of relative water content (also provide the equation)

Line 305 “aged of 4 and 5 years”
Line 325; 327: check throughout the entire document for consistency in writing fold (e.g 3-fold)
Line 329 “were 2.39 fold and 2.62 fold of” is the fold higher or lower?

2. 1st review: The abstract needs revising, the problem is stated, but the significance / importance/need of the study is not elaborated under the abstract. See revised sentence under general comments.
Author: We have revised in Abstract as per the comments, see all the changes in Revising version.

2nd Reviewer reply: Abstract has been revised.

3. 1st review: Numbers describing percentage increases in traits (superoxide, MDA, enzymes and hormones) of the most important observations (highlights) must be included under the abstract.
Author: We have added the changes in the major numbers describing percentage changes in traits (superoxide, MDA, enzymes and hormones).

2nd Reviewer reply: Noted.

4. 1st review: Figures are of high quality, but the figure legend for all figures is not clear, it requires more elaboration. Figures must also be merged to reduce the number of figures in the manuscript. Thanks very much for your comments. We have checked the figure legends, and the results in the previous figures have been merged into 3 new figures.

2nd Reviewer reply: Noted.

5. 1st review: The manuscript lacks some experimental data that will support the findings of current study, for example analysis of relative water content and the photosynthetic rate or photosynthetic pigments to support the data obtained under biomass. Osmolyte content analysis of either proline or soluble sugars can shed some light into the water related or osmoregulation capacity of the plants.
Author: Thanks very much for your comments. We have added the data for the relative water content, SPAD-chlorophyll, and new photosynthesis in leaves of E. sibiricus in revised manuscript.

In the future research, we will consider including osmolyte content analysis of either proline or soluble sugars.

2nd Reviewer reply: Noted. Good luck with the future research, because this is an important study, thus it deserves further research.


6. 1st Review: The importance of Elymus sibiricus must be clearly described in abstract.

Author: The importance of Elymus sibiricus have been stated in Abstract (L13-17).

2nd Reviewer reply: Noted.

7. 1st Review: See added sentence on line 13“Elymus sibiricus L., is a perennial herb that has potential to serve as a grassland for livestock farming”. Elymus sibiricus shows rapid and substantial reductions of aboveground biomass and seed yield after 3 or 4 years of growth, and has an accelerated aging process.

Authors: Thanks very much for your comments. We have added a sentence (L13-17).

2nd Reviewer reply: Noted.

8. 1st review: Line 18-19, sounds as if authors only assayed the antioxidant enzymes and hormones, sentence can be revised “Determining the influence of oxidative damage and hormones signaling”. Appeared? Should be deleted.
Author Response: Thanks very much for your comments. We have revised sentence (L18-19) and deleted appeared on line 27-28.

2nd Reviewer reply: Noted.

9. 1st review: What is the meaning of many-fold? Give a number e.g if you want to generalize rather give a value that is close e.g 50-fold, 10-fold lower etc.
Authors: Thanks very much for your comments. We have added the % or fold changes for most of the parameters in revised manuscript.

2nd Reviewer reply: Noted. I advise that authors, use either % or fold, not at the same time. Consistency is important.

10. 1st review: Under the results section, from line 19 to 31 authors should consider revising this section and include % increase or decrease rather than only stating there was a decrease or increase. Please refer to the response to comment 9.

2nd Reviewer reply: Decide on % or fold.

11. 1st review: Line 44: needs revising, by an English editor

Author: We have revised the sentence in line 57-59.

2nd Reviewer reply: Noted.

12. 1st review: Line 54 grammar
Author: We have revised “influences” to “influenced” on line 68.

2nd Reviewer reply: Noted for comments no. 13 to 22.

13. Line 60, ROS should be written in full as is mentioned for the first time. “Reactive oxygen species” (ROS) is written in full (L73).

14. Line 68, abbreviation for superoxide anion is incorrect We have revised “O2•- ” in the manuscript.
Reviewer reply: Well noted.
15. Line 72, spelling error Author: Revised “indicator” to“ indicate” on line 87.

16. Line 74 – 75, incomplete sentence
Author: We have revised the sentence“the role of oxidative stress in plant senescence and aging has been demonstrated especially in annual and biennial speices (Quirino, Normanly & Amasino, 1999) ”on line 89-92.
17. Line 76-80, needs revising as it is not clear. Author: We have revised it in line 95-103.
.
18. Experiments missing: in addition to the relative water content/photosynthesis/ osmolyte content analysis, element analysis experiments of how nutrients are absorbed and distributed, can also shed light into aging of this crop.
Author: We especially appreciate this comment. We mainly focused on the associations of the antioxidant enzymes and plant hormones with the aging process of E. sibiricus in this article. Following the comment, we will include more measures on osmolytes, elements, and nutrient dynamics during the plant aging process in the future research.

19. Clearly indicate where the E. sibiricus seeds were obtained and include other seeds information. We have revised the seeds obtained and other seeds information (L153-156).

20. Line 126 “Leaves of similar parts were collected from the first to third leaves (counting from the tip of each branch) on each of the branches”. Should be changed to leaves showing similar traits were In the revised manuscript, we have corrected it on line149-150.


21. Line 131 should be revised, “in” should be replaced with “during” Author: In the revised manuscript, we have corrected it on line 155.

22. Overall, lines 131 to 135, should be revised to sound much better.
Author: In the revised manuscript, we have revised the sentences on line 154-160.

2nd Reviewer reply: Line 141-147 in the PDF copy, a reference must be provided for the measurement of relative water content (also provide the equation)

23. Under sample processing and assays there should be subheadings same as line 177
o Determination of lipid peroxidation for MDA content
o Determination of antioxidants enzymes
SOD activity
CAT activity
POD activity
Author: In the revised manuscript, we have added subheadings.
Determination of lipid peroxidation for MDA content on L177.
Determination of antioxidants enzymes on line 191
SOD activity on L192
CAT activity on L198
POD activity on L207

2nd Reviewer reply: Noted for comment 23 to 31.

24. Line 212 has spelling error, please revise We have corrected “roost” to “roots”.
Reviewer reply: Well noted.
25. Revise the sentence: Line 217, “The concentration for 5-year-old plants was lower than that for 4- and 8-year old plants at the heading stage” We have revised the sentence to “O2•-generation rates in leaves of 5- and 8-year old plants were decreased by 38.9% and 27.4% respectively compared to those of 4-year old plants (P < 0.05)” on line 273-274.
Reviewer reply: Well noted.
26. Under O2•-generation rate in leaves and roots, and following sections authors should include % for the obtained results as done on the above sections. We have added the % changes L262-275 in manuscript.

27. Revise line 268-269 We have added the data in manuscript on line 326-330.
Reviewer reply: Well noted.
28. Figure 2: “Plant phenotypes of 3- (A) and 4-year old plant (B) and roots of 4-year old plants (C)”.This figure legend is not very clear, it needs to be revised.

Author: We have deleted the Figure 2. The pictures for plants aged 5-8-years were lost accidently.

29. Line 212, Check Spelling of superoxide anion radical.
Author: We have checked spelling of superoxide anion radical in the manuscript.

30. 1st review: There is a lack of explanation and consistency as to why authors only have data for 3, 4 and 7-years old plants in 2018 and 4, 5- and 8-years old plants in 2019. In the beginning of each section, there must be a clear explanation of what to be expected, looking at the section describing the plant phenotype it needs to be explained that only plants of a particular age will be described and why? Same format must apply to other sections as well.
Author: Thanks very much for your comments. Studies on the aging process of perennial plants are long-term experimentations, for example, it was up to at least 8 years for 8-year old Elymus sibiricus. Considering the year-to-year variation in climates, there may need at least 2-3 replicate years for the same age of plants, thus it will be overall 10 years to get 8-year old plants for 2-3 replicated years. The continuousness of the research project and the collections of samples and data over such a long term could be interrupted by the changes in the financial funding, personnel, and the extreme weather, which were beyond of our control in many circumstances. We are trying very hard to find long-term funding for the research on this important plant species for Tibetan grassland ecosystem, and hopefully to obtain the whole-set of results for the plants aged 3, 4, 5, 6, 7, and 8 years in the future.


31. 1st review: It makes it so difficult to understand the mechanisms behind aging in these plants if some results are missing. I suggest that the focus must be on plants whose data is available for both years 2018 and 2019. Again, what about age 6 plants?
Author: Please refer to our response to comment 30.

32. 1st review: What are the reasons for the 4 year plant having low superoxide in 2018, than the plant in 2019? Explain.
Author: In year 2018, 4-year old plants not only had lower O2•-generation rates but also lower MDA content. It is interesting to note that 4-year old plants also had lower ABA content and higher SPAD- Chlorophyll Value. It could be hypothesized that chloroplasts play a role in plant age-induced oxidative stress, and supporting further the role of ABA and oxidative stress in the regulation of aging in perennials [this is a good statement, but authors to read my comment carefully below].

2nd Reviewer reply: Since the focus here is age, both plants are 4-years old, however sampled in different years, (2018 and 2019), so unless there were other influences such as the climate (weather, high temperatures or rainfall), it is expected that the data obtained for these plants should not vary at a higher degree. Thus any environmental influences must be mentioned. And references to support this finding is important.

33. My analysis of Figure 3: Younger plants are generating higher ROS than older plants e.g 3-year in 2018 and 4-year in 2019. Why? Is this what the authors are expecting?
.
Thanks very much for your questions. What we are expected was the younger plants should have lower ROS than older plants. However, the opposite results were obtained. Compared 2019 to 2018, the generation rates of O2•- in the plot 1 (3-year old in 2018 and 4-year old in 2019) was 26.9%, plot 2 (4-year old in 2018 and 5-year old in 2019) was 26.6% and plot3 (7-year old in 2018 and 8-year old in 2019) was 184.6%. The results showed that the effect of planting age on the superoxide anion radical generation rate of the leaves of the old wheat was greater than that of the younger plants.

2nd Reviewer reply: Noted. Authors to provide references to support these findings in the main manuscript.

34. Lipid peroxidation is observed at older plants 7-year (2018) and 5-year and 8- year (2019) in the leaves, there is a little bit of correlation. Under discussion, authors should provide more clear scientific explanations rather than the influence of environmental factors. We have added some explanations in discussion section L443-456).

Reviewer reply: Noted for comment 34 to 45.

35. Line 240: Change subheading to: Antioxidant enzymatic activity of Elymus sibiricus We have changed the subheading on line 297.

36.1st review: Add this statement “The antioxidant activity of Elymus sibiricus was determined in both the leaves and roots, by assaying the activities of SOD, POD and catalase (Figure)”.
Authors: We have add the statement “The antioxidant activity of Elymus sibiricus was determined in both the leaves and roots, by assaying the activities of SOD, POD and catalase (Figure 3) on L292-293) and also add the statement “The Endogenous hormones of Elymus sibiricus was determined in both the leaves and roots, by assaying the concentrations of IAA, ZT, GA and ABA (Figure 4) on L333-334.

37. 1st review: The article has too many figures, combine the following, Figure 3: Superoxide and MDA contents, this is because they both represent oxidative damage, the subheading can be changed to Markers of oxidative damage or Oxidative stress markers.
Authors: Thanks very much for your suggestions. We have merged the previous figures into 3 figures in the revised manuscript: the O2·- generation rate and MDA content(Fig. 2), antioxidant enzyme activities (Fig. 3), and plant hormones (Fig. 4).


38. 1st review: All enzymes; SOD, POD and CAT on a single figure as SOD (A&B), POD (C&D), CAT (E&F),.
Author: Please refer to response to comment 37.

39. 1st review: Since there is a decline in the antioxidant enzyme activity with an increase with plant age, lipid peroxidation results also showed that MDA contents increased with increasing plant age. This is an interesting findings, I therefore suggest that a statistical correlation analysis must be conducted for all the studied traits to fully understand this aging.
Author: Thanks very much for your suggestions. We have used two-way ANOVA test to show the effects of the plant ages, growth stages, and their interactions on antioxidant activity and endogenous hormone of Elymus sibiricus (Table 4).


40. 1st review: Combine the following figures for the hormones: IAA (A&B) and ABA (C&D); ZT(A&B) and GA (C&D).
Author: Thanks very much for your suggestions. We have combined all hormones into Fig. 4.

.
41. 1st review: The discussion needs to be revised, it must link clearly to the results, e.g The results in this study showed that the SOD activity appeared (Figure),
Author: We have added figure or table in the appropriate locations.


42. 1st review: Line 398, what do you mean by many fold? and consistency is important under the biomass, authors used percentage increase, it should remain the same.
Author: Thank you very much for your comments. Our purpose is to illustrate all the hormones in roots were significant lower than in leaves. There are too many data we had shown in results section in different paragraph.

2nd Reviewer reply: Noted, but Decide on % or fold.
43. 1st review: Authors must interpret the results properly Line 300-304 must be revised.
Author: We have described the ZT content in detail on line 362-366.

44. 1st review: Based on the content of MDA it is clear that the plants were under stress, this is also corroborated by a high increase in antioxidant enzyme activities (check the normal activity of antioxidant enzymes in different ages of plants). This section must be thoroughly discussed.

Author: We have discussed that with the increase of plant aging, the activities of SOD, POD and CAT of antioxidant system decreased, while the content of MDA increased, indicating that the scavenging ability of antioxidant system on reactive oxygen species was weakened, the degree of membrane lipid peroxidation was deepened, and the content of MDA increased.


45.1st review: For Table 4: instead of having the p values, include either a correlation table or correlation figure, it is an easy way to show the relationship between the studied traits in explaining the hypothesis.
Author: We have added some analysis determine the effects of the plant ages, growth stages, and their interactions on antioxidant activity and endogenous hormone of Elymus sibiricus in Table 4.


Added table: Correlations of the parameters between leaves and roots of E. sibiricus at the jointing and heading stages within given year. Data are not presented in the manuscript.

[2nd Reviewer reply: Thank you to the authors for providing the correlation table, Since there is Table 4 in the main manuscript that has been updated, which also indicate the interaction and relationship between the different studied traits, I suggest that authors include this table under supplementary data.]

---

## Round 0.3 · accepted · Accept

Many thanks for your positive attitude towards the comments of reviewers. I think this article is ready for publication.